# EPISTEMOLOGICAL BIAS AS A MEANS FOR THE AUTOMATED DETECTION OF INJUSTICES IN TEXT

## ABSTRACT

Injustice occurs when someone experiences unfair treatment or their rights are violated. The automated identification of injustice in text has received little attention, due in part to the fact that underlying stereotypes are rarely explicitly stated and that instances often occur unconsciously due to the pervasive nature of prejudice in society. Here, we leverage the combined use of a fine-tuned BERT-based bias detection model, two stereotype detection models, and a lexicon-based approach to show that epistemological biases (i.e., words, which presupposes, entails, asserts, hedges, or boosts text to erode or assert a person's capacity as a knower) can assist with the automatic detection of injustice in text. The news media has many instances of injustice (i.e. discriminatory narratives), thus it is our use case here.

## 1 INTRODUCTION

The most basic duty of the media is knowledge sharing. Yet, the tool necessary for wide-spread knowledge sharing is influence. With this influence, the media is able to shape how one will understand the intricacy of a story-line in a news story with little effort. This often results in the use of epistemological biases which involves propositions that are presupposed, entailed, asserted, hedged, or boosted in text (Recasens et al., 2013) to erode or assert a person's capacity as a knower, leading to framing issues and injustice within the text. Particularly, some of these word choices lead to testimonial injustice and what we define as character injustice. Character assassination is "the deliberate destruction of an individual's reputation or credibility" (Icks et al., 2019). This often leads to character injustice, which is an unjustified attack on a person's character that results in an unfair criticism or inaccurate representation of them. Testimonial injustice occurs when modified believability is assigned to the statement of a subject based on widely known stereotypes (Fricker, 2007). Both of these types of injustices can lead to affirming or perpetuating stereotypes concerning the subject. Though it has always been a harsh reality with various ringing consequences, in recent years we have publicly witnessed how the affirmation of stereotypes can lead to physical violence, prejudice, and negative self-image [(Harrison & Esqueda, 1999), (Gover et al., 2020), (Kuykendall, 1989)]. These experiences are harmful and dangerous, explicitly for the victims but for all members of our society. We also consider framing injustice. According to (Entman, 2007) framing bias happens when the use of subjective, one-sided words that reveals the stance of an author occurs. Which means that an individual's choice from a set of options is influenced more by *how the information is worded*, rather than by the information itself. We recognise that news content can be positively or negatively framed to influence the narratives. In this paper, we aim to show how negative cases, when it affects particular subjects or individuals, can lead to framing injustice occurring toward those subjects.

In this work, we seek to make room for subjects of text, even in creative writing, to not have their credibility shot or character assumed due to well-known stereotypes which are harmful and unfounded. Our proposed framework will be used to detect character, testimonial, and framing injustices. The framework includes a fine-tuned BERT model based on work from (Pryzant et al., 2020b) to automatically tag words associated with epistemological bias from an input text, use the (Kwon & Gopalan, 2021) CO-STAR model and (Sap et al., 2020) Social-Bias Frames to find stereotypes and the concepts of those stereotypes associated with the input text, and show when the tagged words (associated with some epistemological bias) or less credibility of a person are correlated with a stereotype which causes injustice. Though we could use examples from various fields (e.g. politics, marketing, medicine, etc...), we will use news media as a use-case throughout this work. Thus, we present the following contributions:

1. We develop a novel framework that uses the results of 3 models to detect character, testimonial, and framing injustices in News Media.

2. We produce a fine-tuned tagger model to automatically detect epistemological bias.

3. We develop a User-Interface for journalists and editors to submit text to and receive output and explanations surrounding the tagged word, referred to in this work as tagger-UI.

4. We produce empirical evidence showing how epistemological bias can translate to injustices.

A goal of this work is to give journalists and editors a tool which will help them easily and quickly learn to avoid character, testimonial, and framing injustices in their work. This will be accomplished by showing users which words they use that produce epistemological bias, showing them the potential stereotype associated with the tagged words and text, offering the user explainability with the help of the stereotype concepts as defined by (Kwon & Gopalan, 2021), and offering the user resources to reference literature on the particular epistemological bias type(s) identified in their input text.

## 2 BACKGROUND

Many works have established it is difficult for the common person to identify a biased word in a sentence and establish the need for computational agents to take on this charge. Section 2.1, 4.1, and 6.1 and Table 8 of (Pryzant et al., 2020b) shows humans have low ability to detect bias and show humans perform worse than their detection model. Recasens et al. (2013) shows in Table 4 that the accuracy of Humans annotators on AMT (amazon mechanical turk) was not more than 37.39% for a single detected biased word. The difficulty arises due to us holding our own biases as facts and lack of education on sentence construction. However, there are specific word choices which are epistemologically biased and can lead to injustices occurring. We will focus on these word choices throughout this work.

Great efforts have been put towards identifying potential words in text materials that could encourage epistemological bias [(Recasens et al., 2013), (Hube & Fetahu, 2019), (Pryzant et al., 2020b)]. Following on from the work of (Pryzant et al., 2020b) we fine-tuned their tagger model to automatically tag words associated with epistemological bias from an input text. Identifying words which cause epistemological bias is a step towards awareness of social harms. This begs the question, what do we do with our new found knowledge and awareness? What kinds of implications does our use of these words impact society? What communities are affected by these word choices? These are the questions we explore in this work.

Authors Kwon & Gopalan (2021) have trained a model to detect widely-known stereotypes and the concept of those stereotypes in text materials. We leverage the results of their CO-STAR model and the Social Bias Frames model (Sap et al., 2020) to offer some explainability of the word choices by the model. Associating a particular text with a stereotype and the concept of that stereotype is a critical step towards awareness of social harms that might cause character injustice to a particular individual or group. Lack of identifying the words in a sentence which imply and promote these stereotypes leaves us with the undirected burden and question of: how can we address these harms? This will be further discussed in the methods section of the paper.

Beach et al. (2021) identify words in text that cause testimonial injustice in medical records of Black patients. Detecting such testimonial injustice is helpful in seeing the unjust realities of our society. They conclude the testimonial injustice that persists in these medical records has a high potential of causing disparity in the quality of health care for Black patients, which correlates with findings that Black patients receive a worse quality of health care (Odonkor et al., 2021). Identifying testimonial injustice in text materials is vital to creating an environment of accountability. For accountability to be applied we must include education, which is often unexplored or left up to the user. Many users do not know where to find resources for such things, thus we provide them in our framework.

Raza et al. (2022) developed a pipeline which takes in news articles, detects and masks words that are biased, and suggests words with more neutral text. Whilst the pipeline and library designed by the authors are very good and useful, they however do not consider the linguistic and epistemological bias features as discussed in (Recasens et al., 2013) and used by (Pryzant et al., 2020b). Unlike

our work discussed in this paper, their pipeline produces no way of distinguishing and highlighting epistemological bias types of any identified potential biased word in the sentence. Also, their framework does not attempt to relate potential stereotypes and stereotype concepts that might be associated with causing injustice to a person or group.

## 3 METHODOLOGY

Hamborg et al. (2019) concluded that various forms of media bias is already analysed in the Social Sciences field and can be implemented in an automated fashion by primarily using Natural Language Processing (NLP) techniques. We leverage research methods using NLP and deep learning whilst also using analysis concepts by researchers from social sciences, who have studied media bias for decades.

We propose a novel technical framework (Figure 1) which includes NLP models for detecting potential epistemological biased words and potential stereotypes along with their concepts, which we semantically link to a given sentence, in our case, a news article headline. We do this with an aim of showing how the automated detection of epistemological bias can help us quickly find injustices in text. This framework is made up of, but not limited to, the attributes discussed here.

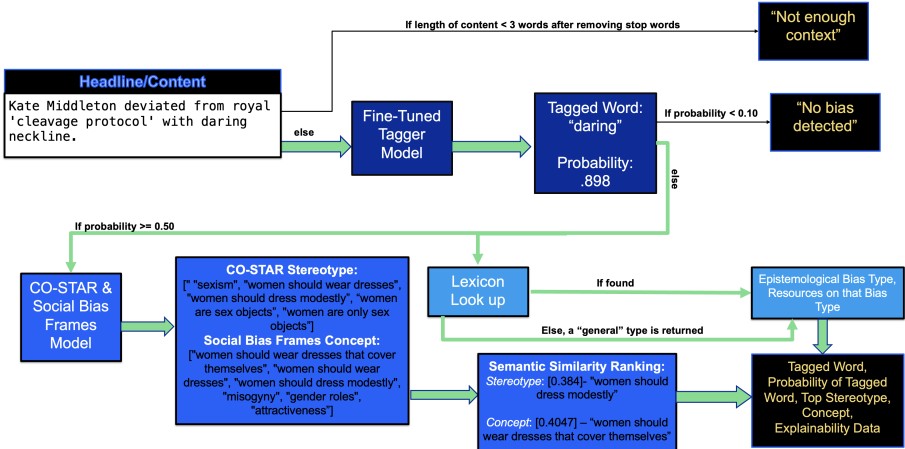

Figure 1: Testimonial Injustice Technical Framework

Figure 2, shows the steps required to scale the framework to analyse multiple headlines.

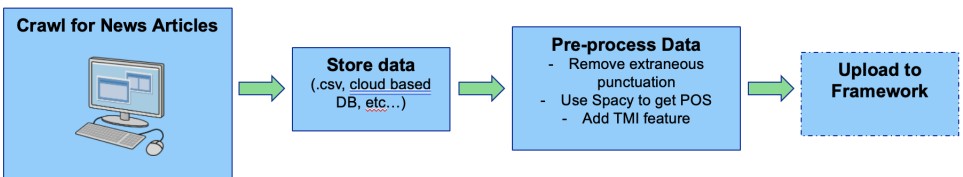

Figure 2: Process of Getting Data to the Framework

### 3.1 TAGGER MODEL

We leverage the detection component of the modular model as discussed in (Pryzant et al., 2020b) and refer to this as the *tagger model* for the rest of this paper.

The tagger model (see Figure 6 in the Appendix) takes as input sentence(s) (e.g. a headline text, a collection of headlines, etc...), and predicts the probability of each word in the sentence(s) been biased (see Equation 1 and Figure 1). Then it returns the word with the highest probability as the

tagged biased word.

$$P_i = \sigma(b_i W^b + e_i W^e + b) \tag{1}$$

Where $b_i \in R^b$ is a word's $w_i^s$ semantic meaning and $e_i$ are the experts features as proposed by Recasens et al. (2013) and based on Equation 2.

$$e_i = R_e LU(f_i W^{in}) \tag{2}$$

The detection model itself is a BERT-based neural sequence tagging model that has been fine-tuned to include the expert linguistic and epistemological bias features from (Recasens et al., 2013). An example of the expert feature as listed in Table 3 of (Recasens et al., 2013) is the *part of speech (POS)* tag of each word in the sentence. Another example of an expert feature from vetted experts in linguistics is *"assertive verbs"*. The model leverages the semantic meaning of each word in the given sentence via the BERT (Kenton & Toutanova, 2019) contextualised word vector. We extend the detection model further by including a feature on whether or not the sentence contained too much information (TMI). We hypothesised here that - the TMI information can be considered from an English linguistic standpoint to be a sentence instance which has more than 2 describing words (i.e. adjectives and adverbs) in it that do not add to its understanding but seeks to cloud the judgement of the readers.We created methods in python which makes use of the core NLP dependency tree and a tree traversal algorithm to go through sentences starting from the root node, then count the number of adjectives and adverbs in the sentence to determine if there is "TMI" or "no TMI" based on the hypothesis. TMI is not a known indicator of epistemological bias, but we acknowledge it might contribute to causing doubt which leads to injustice. This is why we introduce this new feature to see if it will be a contributing feature for the tagger model to detect epistemological biases. We conducted an ablation study to see the effects of adding this feature, whilst training the tagger model in Section 4.1.

## 3.2 CO-STAR MODEL AND SOCIAL BIAS FRAMES (SBF)

The CO-STAR (COnceptualisation of Stereotypes for Analysis and Reasoning) framework allows input from the user and generates outputs of stereotypes and stereotype concepts. The Social Bias Frames model generates and classifies stereotypes associated with an inputted text. These outputs are quite simple to understand but have a very complex history. The accuracy of the baseline SBF model was analyzed by looking for the demographic group the statement targeted and the implied stereotype from the statement. The accuracy was measured with BLEU-2 (group - 83.2%, stereotype - 68.2%) and Rouge-L (group - 49.9%, stereotype - 43.5%). The authors of the CO-STAR model manually evaluated their model, but did not specify the results of their evaluation. However, we manually analyzed how each of these models performed on our sentences and found them to generate stereotypes which are well fitted to the sentences submitted.

Since testimonial injustice occurs because of widely known stereotypes, the outputs of these models will help us inform our users of any potential stereotypes that are exacerbated by their inputted text. The potential stereotype can also be used to determine if a person's character has been unjustly targeted. This is because stereotypes are not based on the actual truth of the particular person they are attached to and stigmatises the individual. Their presence amidst character assassination is evidence of character injustice. Stereotype concepts will offer explainability as to why a certain word was tagged as being epistemologically biased.

We submit news article headlines to the CO-STAR and SBF models and receive an output of 6 potential stereotypes and 3 stereotype concepts the sentence is related to. Semantic similarity describes how closely related two items (e.g. 2 words or 2 sentences) are in terms of meaning. We use semantic similarity to encode the list of generated sentences and the original headline sentence, then using distance metrics on the encoded vectors, e.g. using a cosine similarity metric from sentence transformers, to get the distance scores. We rank the outputs based on their semantic similarity to the headlines. The stereotype and stereotype concept most correlated with the sentence is output to the user as the potential stereotype and concept which casts doubt on the subject of the inputted text.

## 3.3 LEXICON LOOKUP

Once the tagger model has returned the top tagged word, we then proceed to automatically look up and semantically search the tagged word in the epistemological lexicons from social sciences - to

discover the epistemological bias types it is associated with. These lexicons are from the collection of datasets we discuss in the dataset section 3.5. When a tagged word is not found in the lexicons, we lemmatize said word (considering its context) to find its base word and find the stem word (removing the prefixes and suffixes). We then search for the lemmatized and stemmed words in the lexicons.

### 3.4 INTERACTIVE INTERFACE FOR LEARNING ABOUT BIAS

We propose the use of an interactive interface for editors of media content to learn more about and mitigate any potential bias types and associated stereotypes, prior to them publishing it.

During the analysis, we first leverage the nltk python library to remove any stop words (i.e. a, the, etc..). We then check the sentence length. If the sentence is less than 3 words, we determine there is not enough context to analyze the text for potential stereotypes. However, if we have enough context, the content is sent to the epistemological tagger model to find the word with the highest probability of bias (see Figure 7 in the Appendix). The conditions from the framework (Figure 1) are then implemented.

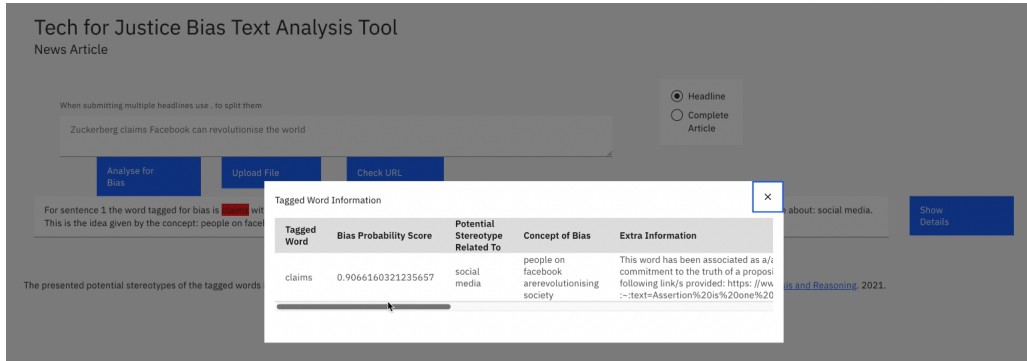

Figure 3: The interactive interface showing the tagged words, their associated probability for bias, the potential stereotype and stereotype concept and some explainable data created from the lexicon search.

We display the types of associated epistemological bias detected to the user and provide links to resources which explains more on the specific types of biases as shown in Figure 3. See examples of these outputs in table 7 and table 6 of the appendix section.

| Sentence No. | Headline | Subject |
|---|---|---|
| 1 | Meghan Markle spent a staggering £38,000 on her clothes for a charity trip | Meghan |
| 2 | Kate Middletons £100,000 Astonishing value of the dress that won a Prince's heart (and has hung in a wardrobe for eight years) | Kate |
| 3 | Meghan Markles beloved avocado linked to human rights abuse and drought, millennial shame | Meghan |
| 4 | Kates morning sickness cure? Prince William gifted with an avocado for pregnant Duchess | Kate |

Table 1: Example headline sentences used. For a full list of the examples, see A

### 3.5 DATASET SOURCE USED

We leverage and used the bias data corpus[1] as created by (Pryzant et al., 2020b). It contains a Wikipedia neutrality corpus showing Wikipedia articles that were annotated for neutrality by editors on Wikipedia, who adhere to NPOV (neutrality point of view) guidelines [2].

Lexicons used in the lexicon lookup were from social science research and collated in (Pryzant et al., 2020a). We compiled these lexicons into one large dictionary and included some metadata about the

---

[1]http://nlp.stanford.edu/projects/bias/bias_data.zip
[2]https://en.wikipedia.org/wiki/Wikipedia:Neutral_point_of_view

lexicon i.e. source, creators, resources about the epistemological bias type, etc. Solely using an epistemological lexicon look up directly to find the biased words will be limited in its performance, for various reasons e.g. distributional shift, lexicons cannot scale well enough without requiring regular manual auditing, updating of the lexicon databases - when newer forms of subtle biased words arise, etc. There are various studies which show the limitations of using just a lexicon based approach alone, for example in (Cryan et al., 2020), the authors discuss on page 8 and show in Table 5 how lexicon approaches perform less accurately than an end to end approach using BERT to detect gender based stereotypes. Therefore, it is beneficial to not solely lean on lexicons and we include it as a contributor in detecting injustices in our framework.

When it comes to Meghan Markle and Kate Middleton, the media depicts several aspects of their lives which they share differently and often can be reflective of framing or character injustices[3]. We use actual headlines (as seen in Table 1) from such depictions in our comparative test to further illustrate the detection and harms of these injustices. The results of this comparative test are further discussed in our results section (Section 4).

## 4 RESULTS AND DISCUSSIONS

### 4.1 TAGGER MODEL ABLATION STUDY

We performed an ablation to analyse the fine-tuned tagger model performance when there is no expert feature as proposed by Recasens et al. (2013)and when they are included. Part of the training experiment was also to determine if there are any benefits from including the TMI feature into the input data. We carry out the training on 23,000 training samples from the bias-data training set and 700 validation and 1,000 test samples. Training was done using 4 CPUs and 1 GPU. We used a learning rate of 3e-5 and initially train for 10 epochs.

| Tagger Model Ablation Experiments | | |
| --- | --- | --- |
| Kind | Evaluation Accuracy(%) | Evaluation Loss |
| basic | 72.54 | 0.0758 |
| | 72.39 | 0.0766 |
| | 73.13 | 0.0851 |
| + expert features | 75.04 | 0.0745 |
| | **74.16** | **0.0734** |
| | 72.83 | 0.0867 |
| + tmi | 74.79 | 0.0730 |
| | *74.63 | *0.0703 |
| | 74.63 | 0.0822 |

Table 2: Ablation study results showing values for the first 3 epochs. Note: A high score is better for *accuracy* and a lower score is better for *loss*. The numbers highlighted in bold are the values for when the evaluation loss was less than the previous step.

As shown in Table 2 and Figure 4, we observe a lower training loss value between epoch 3 & 6 during the initial training using the dataset which included the TMI feature - as compared to when we do not. TMI also gives a better evaluation accuracy and evaluation loss as compared to the others before any overfitting is observed (which causes the evaluation loss to exponentially increase from about epoch 3). This overfitting occurs as a result of using too many epochs during the initial training of the tagger model. When fine-tuning a BERT model, a good practice as suggested by the authors of Bert is to use between 2 and 4 epochs. In light of this, and from the observations made from the initial training experiments, we save and use as the best point the tagger model after epoch 2.

### 4.2 COMPARATIVE TEST

We performed a comparative test to illustrate how our framework shows injustices e.g. character and framing injustice. Particularly, we capture the type of epistemological bias attached to the tagged word in each entry, observe if a relevant stereotype is associated with the inputs, and observe the depiction of different subjects. In this study, we look at Meghan Markle and Kate Middleton

---

[3]https://www.buzzfeednews.com/article/ellievhall/meghan-markle-kate-middleton-double-standards-royal

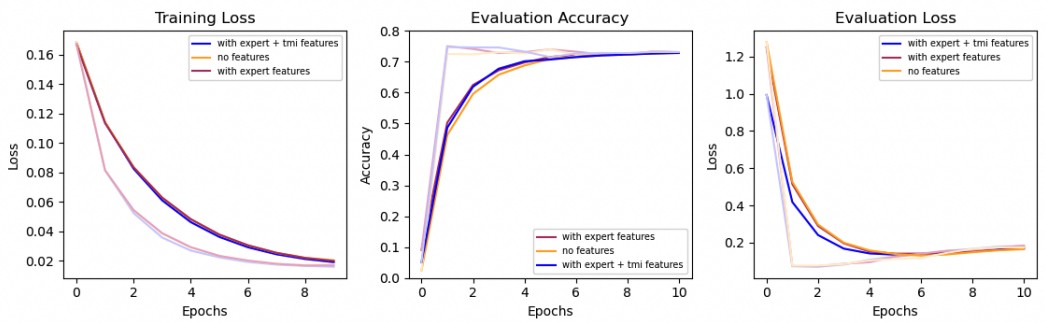

Figure 4: Training and Validation Results for the Tagger Model using 10 epochs. ***no features*** = the expert features as mentioned in section 3.1 was not added, ***with expert features*** = combine the expert features as mentioned in section 3.1 and ***with expert + tmi features*** = combine the expert features whilst using the data samples containing TMI information class *no tmi* or *tmi*

as our subjects of interest. Meghan Markle is an American, Black woman member of the British royal family, by marriage, who has recently resigned from her royal duties due to abuse from the media and royal family (e.g. the Firm). Kate Middleton is a British, White woman member of the British royal family, by marriage, who has recently been promoted to Princess of Wales. Even before Meghan Markle had resigned from her royal duties, the media has sought to minimize her experience and diminish her comments and character. We often see instances of Meghan as the subject of an article and the media using sarcasm and criticism towards her. Such acts have led readers to having tainted images of Meghan and even not believing her statements or actions, thus testimonial injustices. We also see this leading to her character being questioned, thus character injustice. Yet, for the same and similar topics of concern, media members speak charmingly about Kate, thus leading to framing injustices shown. It is important to note here, that Kate Middleton and Meghan Markle shared similar positions, interests, and abilities. With this, we see them as good subjects to observe in our comparative test.

| No | Subject | Tagged Bias Word | Potential Bias Type | Sentence Sentiment |
|----|---------|------------------|---------------------|--------------------|
| 1 | Meghan | ['staggering'] | ['subjectives'] | negative |
| 2 | Kate | ['astonishing'] | ['positive', 'subjectives'] | neutral |
| 3 | Meghan | ['beloved'] | ['positive', 'subjectives'] | negative |
| 4 | Kate | ['gifted'] | ['positive', 'subjectives'] | neutral |

Table 3: A subset of output result. (It shows the words tagged as potentially biased, the lexicon that type of bias is associated with, and the overall sentence sentiment for each headline sentence depicted in table 5 & full list of the output results see table 6). For a full list of columns, please refer to the appendix A

In Figure 5, we plotted results of an experiment where we ran 10 article headlines of Meghan Markle and 10 articles of Kate Middleton through our framework. Each article was chosen because it discusses a topic which is common between each of the two subjects and allows us to see the differences in how the two subjects are spoken about concerning that topic. It has been acknowledged by several outlets that these particular articles which we have mentioned are unjust[3]. Majority of these articles are seen in the aforementioned articles to show these comparisons, but also a few additional articles were hand chosen by our team. You can see the headlines for each subject in Table 1. The complete result output and table column descriptions are given in Appendix A. In this plot, we capture the subject of the headline (Meghan/Kate), sentiment of the entire headline sentence, and the associated epistemological bias types for each headline as shown in table 3. The light blue circles represent the overall sentiment of each headline, we will refer to them as the sentiment headline level. We use the 3 categories of positive, neutral, and negative sentiments. Within each sentiment headline level, we capture the subject of the headline, we will call this the subject level. Note that in Figure 5, Kate takes up the entire positive sentiment space, there were no examples that were detected to have positive sentiments for Meghan. Within each subject level, we capture the epistemological bias types associated with the tagged words in the headlines for each subject; we will refer to this as the bias-type level. This illustration shows us how much space is filled by each subject.

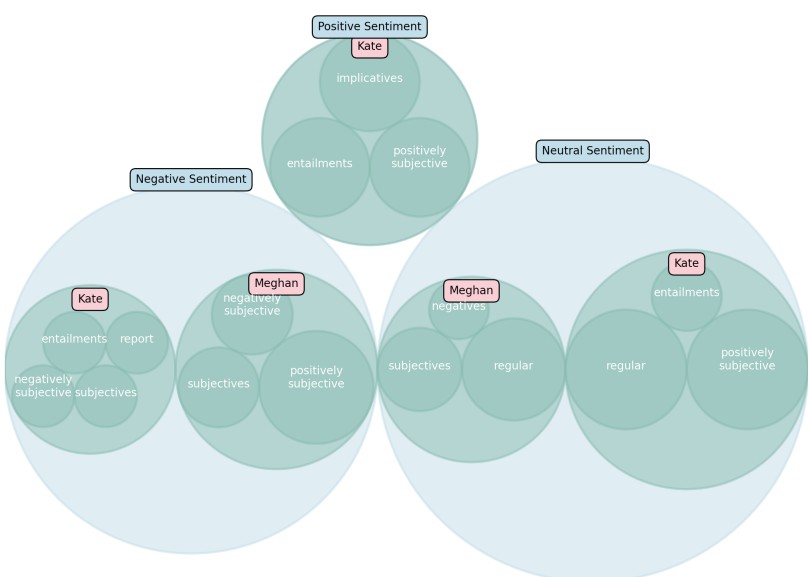

Figure 5: Epistemological Bias Types - Comparative Test

From this plot, we can see the headline sentiment level, the negative sentiment circle is mostly filled with articles about Meghan and the most common bias type is *positively subjective*. The intuition here is that many articles about Meghan contain sarcasm, though this is not the only indicator. There was one article about Kate which had negative sentiment and each of the bias types in her circle were identified in the headline, this is why they are all equal in size. In looking at the headline sentiment level, the positive sentiment circle is completely filled with articles about Kate and the most common bias type is *implicatives*. The intuition here is that, in articles which have common topics between Meghan and Kate, only Kate is talked about in a suggestively positive way. In looking at the headline sentiment level, the neutral sentiment circle is mostly filled with articles about Kate and the most common bias type is *positively subjective* and *regular*. Recall, regular is the epistemological bias type assigned to detected words which do not appear, nor do similar words appear in the lexicon lookup. We can also observe, the articles which appear for Meghan with neutral sentiment are *negatively subjective* in nature, informing us they likely are not sarcastic but have negative suggestions. For Kate, she is talked about in a *suggestively positive* way, even in the neutral sentiments. It is clear from this experiment when there is a topic that Meghan and Kate share - the media uses words for Meghan which are sarcastic and cause framing and character injustices to her as a subject. Both of which can contribute to someone taking Meghan less seriously, thus causing testimonial injustice.

| No. | Subject | Stereotype(S) | Distance(S) | Stereotype Concept(SC) | Distance(CS) |
|---|---|---|---|---|---|
| 1 | Meghan | personal spending habits | 0.3457 | women should spend money on clothes | 0.4914 |
| 2 | Kate | women should be dressed like brides | 0.3259 | women are property | 0.2278 |
| 3 | Meghan | feminism | 0.3663 | sexism | 0.2523 |
| 6 | Meghan | family law | 0.3244 | arab folks are not protected | 0.2951 |
| 10 | Kate | women should wear dresses | 0.3131 | women should wear dresses | 0.3131 |
| 18 | Meghan | royalty housekeeping are gold diggers | 0.3741 | royalty house queens are gold diggers | 0.3657 |
| 20 | Meghan | women should dress modestly | 0.3840 | women should wear dresses that cover themselves | 0.4047 |

Table 4: Table showing the potential top ranked stereotype and stereotype concept (only those with a semantic distance $> 0.3$ to the headline are shown in this table. The No. column represents the headline number from Table 5). For a full list of result see Table 7 in Appendix A.

Looking at the potential stereotypes generated with a semantic similarity distance greater than 0.3 to the headline, as shown in table 4 (see appendix table 7 for the full table), we observe that 5 out of 7 of such entries related to Meghan as the subject and only 2 related to Kate. More interestingly, we

observed that one of the potential stereotypes relating to Meghan (*personal spending habits*) aimed at a personal attribute. Which can be seen as an indication of a potential character assassination, thus character injustice.

### 4.3 POTENTIAL ISSUES ELEVATED DUE TO THE PROPOSAL AND MEANS TO RESOLVE THEM

A critical area for bias in systems and models' design often stem from a given human's intrinsic biases. They are usually a reflection of ourselves. One possible solution to this problem is to ensure that a diverse group of individuals are involved from the inception of the solutions design to the testing phase of such technical solutions—this is one reason why we gathered a diverse group of authors to be involved in the discussions presented in this paper.

A second area for bias surrounds the definition of terms and assumptions biases. A way we attempt to resolve this is by considering the existing consensus definitions of epistemological bias from the social sciences, where media bias has been studied for decades.

A third potential area for bias can occur in the training and validation datasets, as well as in the models used for implementing the proposed solution. For example, the sourced datasets do not contain enough semantic information that is very reflective of the injustices considered. One possible way to combat this is to investigate ways to build more robust training datasets or leverage models that do not require lots of training examples to generalise properly.

Broader problems may arise in any situation where technology is naively applied to solve a societal issue. As envisaged, our framework should be applied as a means to help people working in the media improve their output with respect to bias and injustices. However, as warned by Goodhart's law (Manheim & Garrabrant, 2018), if the measures and metrics suggested here become targets, they will cease to be useful. For example, in situations where experts deliberately bias their content the tool can become beneficial to the readers instead, so that they are aware of potential biases when reading an article. However, a main purpose of our proposed tool and concept is to help journalists who are aware that they might use biased terms but do so unintentionally. On the other hand, we cannot control the adoption of our tool. It will help the editors who are checking for bias using manual means to quickly detect such biases.

## 5 CONCLUSIONS

We propose a framework which uses an epistemological tagging model, a stereotype detection model and social bias frame model, combined with semantic searching and lexicon lookup of epistemological biased word to detect character, testimonial, and framing injustice. These forms of injustices are often subtle and hard to quickly detect or be aware of. We also provide empirical evidence as a justification for using this framework in media settings to detect such injustices. Further, we propose an interactive interface which can aide editors and journalists of news text content by automatically detecting and explaining potential bias types and injustices that might be present in their content. We anticipate this interactive interface will encourage them to take necessary, preventative steps in avoiding unjust acts before their content is released.

## 6 FUTURE WORK

- More Comparative Test data:- Our goal was to conduct an empirical qualitative research study which shows the framework (see Figure 1) can be implemented to detect injustices. However, we recognise that a limitation of the comparative test in this paper is in using only 20 headlines, and we hope to show the scalability of the work in future studies.
- Learning Better Patterns for Testimonial injustice:- Identifying a single word which potentially cause bias in a given sentence is only a start. We acknowledge that multiple words or phrases in a given sentence might be the culprit and the ability to tag multiple words or phrases will take this work further.
- Veridicality Assessment:- We intend to incorporate veridicality assessment in our framework to assess if a statement is actually factual, which will allow for a more accurate analysis.

## 7 REPRODUCIBILITY STATEMENT

We discuss how we leverage the models, our framework and the outputs of the interactive interface of this work in section 3. To access the CO-STAR and SBF models discussed in this work we direct you to the GitHub repository (Kwon & Gopalan, 2021) have created `https://github.com/kwonathan/CO-STAR`. The datasets used in our work are discussed in section 3.5. Each dataset is publicly available, except one which is proprietary and used to tune the tagger model. However, the tagger model can be fine-tuned to a dataset which is relevant, ethically gathered, and appropriately constructed.

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

## A  APPENDIX

In Figure 6, we have a diagram showing how the tagger model computes logits $y_i$ using discrete feature $f_i$ and BERT embedding $b_i$ Pryzant et al. (2020b).

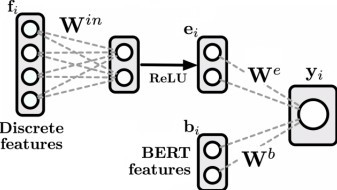

Figure 6: The tagger model computes logits $y_i$ using discrete feature $f_i$ and BERT embedding $b_i$

In Figure 7 we have a screenshot of the UI showing immediately after a user has submitted a text for analyzing. The word suspected of causing injustice is highlighted with the certainty score, an explanation of why the word was tagged. When the user clicks *Show Details* they will see more details as displayed in Figure 3

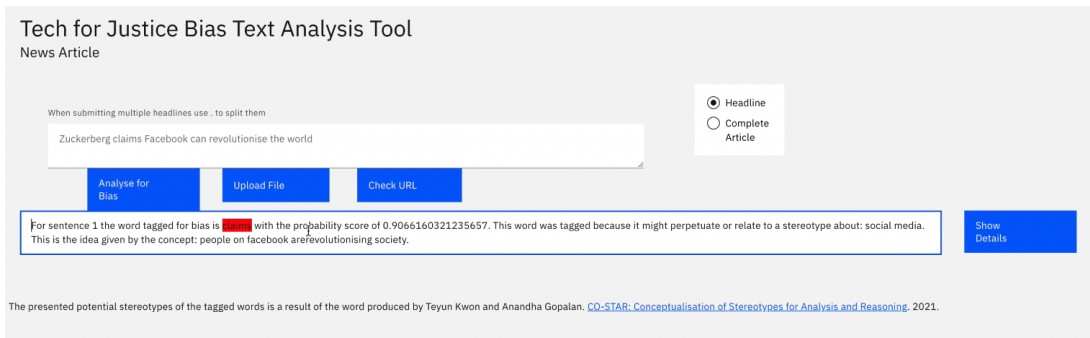

Figure 7: The interactive interface showing the highlighted tagged word and associated information for a sentence.

In Table 5, we have all the headlines used in the comparative study between Meghan Markle and Kate Middleton. Several news outlets have also noted that these articles are used unjustly to depict one person differently than the other on a shared topic, playing on stereotypes about the person.

| Sentence No. | Headline | Subject |
|---|---|---|
| 1 | Meghan Markle spent a staggering £38,000 on her clothes for a charity trip | Meghan |
| 2 | Kate Middletons £100,000 Astonishing value of the dress that won a Prince's heart (and has hung in a wardrobe for eight years) | Kate |
| 3 | Meghan Markles beloved avocado linked to human rights abuse and drought, millennial shame | Meghan |
| 4 | Kates morning sickness cure? Prince William gifted with an avocado for pregnant Duchess | Kate |
| 5 | Kate and William 'packed up the kids' in search of 'privacy' at new Windsor estate | Kate |
| 6 | Why Prince Harry and Meghan Markle's Potential Plan to Protect Their Family is Incredibly Unfair to Archie | Meghan |
| 7 | Prince Harry and Duchess Meghan Think Moving to Canada Will Give Archie a Normal Upbringing | Meghan |
| 8 | Not long to go! Pregnant Kate tenderly cradles her baby bump while wrapping up her royal duties ahead of maternity leave - and William confirms she's due 'any minute now' | Kate |
| 9 | Why can't Meghan Markle keep her hands off her bump? Experts tackle the question that has got the nation talking: Is it pride, vanity, acting - or a new age bonding technique? | Meghan |
| 10 | Kate Middleton Wore a Bardot Dress to the 'Top Gun' Premiere | Kate |
| 11 | Kate Middleton's homegrown bouquet of lily of the valley follows royal code | Kate |
| 12 | Royal wedding: How Meghan Markles flowers may have put Princess Charlottes life at risk | Meghan |
| 13 | Duchess Kate reveals her favourite photo of son Prince Louis | Kate |
| 14 | How Meghan and Harry Ripped Up Royal Tradition on Birthday Photos of Archie | Meghan |
| 15 | Kate Middleton Debuts New Sapphire Earrings That Belonged to Princess Diana and Debunks a Rumor! | Kate |
| 16 | Meghan Markle Just Casually Rewore Her $16,500 Royal Wedding Earrings in NYC | Meghan |
| 17 | Kate and Wills Inc: Duke and Duchess secretly set up companies to protect their brand - just like the Beckhams | Kate |
| 18 | A right royal cash in! How Prince Harry and Meghan Markle trademarked over 100 items from hoodies to socks SIX MONTHS before split with monarchy - with new empire worth up to £400m | Meghan |
| 19 | Kate Middleton deviated from royal 'cleavage protocol' with daring neckline | Kate |
| 20 | Meghan Markle Just Wore a Plunging Gown With a Thigh-High Slit on the Red Carpet | Meghan |

Table 5: Comparative Headline Sentences Used

Table 6 in this section shows the full output of tagging each headline sentence used in the comparative test. The `No.` column corresponds to the sentence number as shown in table 1, `Subject` refers to the person subject the headline relates to, `Taggerout Bias` represents the tagged word from the sentence that could potentially be biased, `Taggerout Prob` refers to the probability of the tagged word been biased, `Taggerout in lexicon` refers to the results of checking whether the corresponding tagged word is observed in the lexicon of associated epistemological biased words. This offers some explanability for the potentially biased word. A **True** value for this column indicates that the word is indeed in the lexicon, whilst a **False** value indicates otherwise. `Bias Type` indicates what kind of epistemological bias type is associated with the tagged word. A **regular** bias type word indicates that the tagged word was not found in the epistemological bias lexicon. The `Sentence Sentiment` columns represents the overall sentiment associated with the headline sentence (which can be obtained using any NLP processing tool e.g. spacy).

| No. | Subject | Taggerout Bias | Taggerout Prob | Taggerout in Lexicon | Bias Type | Sentence Sentiment |
|---|---|---|---|---|---|---|
| 1 | Meghan | ['staggering'] | 0.999498 | True | ['subjectives'] | negative |
| 2 | Kate | ['astonishing'] | 0.999342 | True | ['positive', 'subjectives'] | neutral |
| 3 | Meghan | ['beloved'] | 0.997946 | True | ['positive', 'subjectives'] | negative |
| 4 | Kate | ['gifted'] | 0.877285 | True | ['positive', 'subjectives'] | neutral |
| 5 | Kate | ['packed'] | 0.478948 | False | ['regular'] | neutral |
| 6 | Meghan | ['incredibly'] | 0.998493 | True | ['positive', 'subjectives'] | negative |
| 7 | Meghan | ['normal'] | 0.689049 | True | ['subjectives'] | neutral |
| 8 | Kate | ['confirms'] | 0.997673 | True | ['entailments', 'report'] | negative |
| 9 | Meghan | ['vanity'] | 0.599388 | True | ['negative', 'subjectives'] | negative |
| 10 | Kate | ['top'] | 0.933422 | True | ['entailments', 'positive', 'subjectives'] | neutral |
| 11 | Kate | ['homegr'] | 0.820782 | False | ['regular'] | neutral |
| 12 | Meghan | ['royal'] | 0.551708 | False | ['regular'] | neutral |
| 13 | Kate | ['favourite'] | 0.838293 | False | ['regular'] | neutral |
| 14 | Meghan | ['ripped'] | 0.890869 | True | ['negative'] | neutral |
| 15 | Kate | ['bunks'] | 0.758531 | True | ['negative', 'subjectives'] | negative |
| 16 | Meghan | ['casually'] | 0.967407 | False | ['regular'] | neutral |
| 17 | Kate | ['just'] | 0.645452 | True | ['subjectives'] | negative |
| 18 | Meghan | ['markle'] | 0.583964 | False | ['regular'] | neutral |
| 19 | Kate | ['daring'] | 0.856008 | True | ['entailments', 'implicatives', 'positive', 'subjectives'] | positive |
| 20 | Meghan | ['just'] | 0.565576 | True | ['subjectives'] | neutral |

Table 6: Table showing the output of passing each headline through the model that tags the potentially biased word in the given sentence.

In Table 7, the `Stereotype (S)` column refers to the closest potentially associated stereotype. After passing each headline through the CO-STAR and Social Bias Frame models as discussed in section 3.2, we semantically ranked the list of potential stereotype to the headline sentence, and then selected the top ranked potential stereotype and stereotype concept - as seen in the column `Stereotype Concept(SC)`. The `S Distance` and `SC distance` columns shows the semantic similarity distance of the stereotype and stereotype concepts to the headline sentence.

| No. | Stereotype (S) | Distance(S) | Stereotype Concept(SC) | Distance(SC) |
|---|---|---|---|---|
| 1 | personal spending habits | 0.3457 | women should spend money on clothes | 0.4914 |
| 2 | women should be dressed like brides | 0.3259 | women are property | 0.2278 |
| 3 | feminism | 0.3663 | sexism | 0.2523 |
| 4 | british women are marginalized for a joke | 0.2083 | pregnancy | 0.3918 |
| 5 | N/A | N/A | N/A | N/A |
| 6 | family law | 0.3244 | arab folks are not protected | 0.2951 |
| 7 | arab folks should live in a constant state of worry | 0.1618 | racial hierarchy | 0.0999 |
| 8 | gender hierarchy | 0.1178 | women are often subjected to terms like "babies" and "pregnant" | 0.2632 |
| 9 | women are sexual objects | 0.2602 | misogyny | 0.2574 |
| 10 | women should wear dresses | 0.3131 | women should wear dresses | 0.3131 |
| 11 | women are property of men | 0.1226 | women are property | 0.1494 |
| 12 | women are sex objects | 0.1676 | women are vulnerable | 0.2495 |
| 13 | sexual abuse | 0.0746 | saudi arabians are pedophiles | 0.1226 |
| 14 | sexual assault | 0.2091 | royalty house guests are paedophiles | 0.4691 |
| 15 | women are sex objects | 0.1637 | women are property | 0.1504 |
| 16 | black folks want to marry rich people | 0.2266 | racial marriage | 0.0869 |
| 17 | women are property | 0.2026 | women are secret business partners | 0.3732 |
| 18 | royalty housekeeping are gold diggers | 0.3741 | royalty house queens are gold diggers | 0.3657 |
| 19 | sexism | 0.1927 | gender hierarchy | 0.2284 |
| 20 | women should dress modestly | 0.3840 | women should wear dresses that cover themselves | 0.4047 |

Table 7: Table showing the potential top ranked stereotype and stereotype concept. We passed each sentence through the CO-STAR and social bias frames model. After which the potential stereotype and stereotype concepts where semantically ranked to the sentence itself

