# OpenReview forum: "Epistemological Bias As a Means for the Automated Detection of Injustices in News Media"
_ICLR.cc/2023/Conference — Submitted to ICLR 2023_

### Official Review · Reviewer_P7BJ · 2022-10-22

**Confidence:** 3
**Correctness:** 3
**Technical Novelty And Significance:** 2
**Empirical Novelty And Significance:** 2
**Recommendation:** 3

**Clarity, Quality, Novelty And Reproducibility:**

Clarity: The paper is well written with some minor issues. However, a lot of terminology is used that might not be known to the ICLR community. Some more gentle introductions or intuitive examples would have been helpful.
Quality: The qualitative research is good. However, the quantitative part is not sufficient. The paper presents convincing anecdotal evidence, but how this approach works on large sets of non-handpicked headlines is unclear.
Novelty: The concepts presented and computational framework proposed are novel, but not from a technical contribution and not with respect to empirical insights.
Reproducibility: Since most components of the framework are openly accessible, the work is mostly reproducible.

**Strength And Weaknesses:**

+ The text analysis tasks proposed in this paper go beyond most established tasks with respect to the linguistic complexity required to solve the task.
+ The application area is very timely and tackles a pressing societal issue.
+ The authors are aware of the limitations of the work and make them explicit.
- The paper is hard to access from a computer science oriented perspective, since a lot concepts from social sciences are used without definition or intuitive examples.
- The technical contribution is restricted to the conceptual framework. No novel algorithmic contributions are provided, nor quantitative empirical insights.
- While the task of bias detection is being addressed in a more fundamental way than before, the paper misses a larger vision, like using the tool to analyze bias quantitatively on large scale media data.

**Summary Of The Paper:**

The authors propose a framework for epistemological bias detection, by combining different (existing) tools and features from previous work. By doing so, they claim to detect three types of bias: character, testimonial, and framing injustices. This bias is associated to single words, while exploiting context on the sentence level. Once a word is tagged it is linked to an entry in an epistemological lexicons from social sciences. This helps to explain the decision of the system to a user and to associate it to potential stereotypes. The authors see journalists and editors as the main target group of their tool, since it can be used to reduce unwanted bias in their articles.
The model is qualitatively evaluated on a data set with 20 headlines to reveal sentiment and bias type.

**Summary Of The Review:**

I believe the paper's contribution is to push NLP more towards fundamental concepts from social and communication science. However, the lack of algorithmic novelty or statistical empirical evidence on the performance of the model makes it not yet ready for acceptance.

---

> ### Author Response · Authors · 2022-11-16
> **Reply 1: P7BJ**
>
> Thank you for your review. We have addressed your concerns with edits to our paper. We welcome additional feedback and improvements.
>
> - **Thank you for the positive feedback of:** *"The text analysis tasks proposed in this paper go beyond most established tasks with respect to the linguistic complexity required to solve the task."* > **We have further emphasized this point by adding an image of the framework to section 3.**
> - *"The paper is hard to access from a computer science oriented perspective, since a lot concepts from social sciences are used without definition or intuitive examples."* > **We are happy to make writing changes to define the unclear terms referred to. However, we need clarity on which terms we need to provide more definitions for. We can create an appendix with terms | definitions | example. Can you provide some insight here to the terms you found unituitive?**
> - *"The technical contribution is restricted to the conceptual framework. No novel algorithmic contributions are provided, nor quantitative empirical insights."* > **As rightly highlighted by you, our goal for the paper was to contribute a technical, conceptual framework, which showed how algorithms are applied to solving real-world problems in a novel way. We also aimed to carry out and provide a qualitative empirical insight into the problem area discussed, as opposed to a quantitative empirical study. With this, we are sure our paper contributes well to this submission track of Social Aspects of Machine Learning. Additionally, we have updated the paper to add more technical details in sections 3, 3.1, 3.2, and 4.1. Another technical contribution made is the proposed UI. We have updated the paper to add screenshots of the UI in section 3.4 and the Appendix. Our final technical contribution is the fine-tuning of the Tagger Model. We have updated sections 3.1 and 4.1 with more technical details here.**
> - *"While the task of bias detection is being addressed in a more fundamental way than before, the paper misses a larger vision, like using the tool to analyze bias quantitatively on large-scale media data. How this approach works on large sets of non-handpicked headlines is unclear."* > **We acknowledged the limitation of using a few samples of headlines in our evidence and analysis portion to show the task of detecting injustice in Section 6. We would like to highlight when using the tool it can scale up to be used on large-scale media data. It can span large-scale, multiple volumes of news articles which can be collected with an API crawl, stored in a document, and uploaded. We have updated our paper to show this by including the process by which one might add multiple volumes of data in section 3. Please also see the added screenshots of the UI in section 3.4 and the Appendix. Additionally, we have updated the paper to mention other domains in which the tool could be used in section 1. We also updated the Title and abstract to emphasize the news media is only a use case for this work.**
> - *"lack of algorithmic novelty or statistical empirical evidence on the performance of the model"*  > **We have updated sections 3.2 and 4.1 to discuss the accuracy of the models used in this work.**
>
> We thank you for your review and welcome any additional feedback and further review. You will find our updated paper is now uploaded. We ask you to review these updates and consider increasing your support for our paper.

---

### Official Review · Reviewer_zugw · 2022-10-23

**Confidence:** 3
**Clarity, Quality, Novelty And Reproducibility:** The paper was very clear and it shows…
**Correctness:** 3
**Technical Novelty And Significance:** 4
**Empirical Novelty And Significance:** 3
**Recommendation:** 8

**Strength And Weaknesses:**

Strengths:
1. The paper was well written; the flow and arrangements of the sections are excellent
2. The approaches discussed and used are current and detailed to some extent
3. The presentation of the overall result is impressive and relevant

Weaknesses:
1. The framework developed was not shown - a schematic diagram or an algorithm would have sufficed.
2. The approach of the development of the tagger-UI was not presented - what software development methods were employed?
3. The tagger-UI was not presented in the work - a screenshot would have sufficed.
4. No mathematical model was demonstrated to prove the theory behind the entire process.
5. How the sentiments (pos, neg) were derived was not shown clearly

**Summary Of The Paper:**

The work used news media dataset to detect character, testimonial, and framing injustices. The models  employed are fine-tuned BERT model, CO-STAR model and Social-Bias Frames model. The framework is able to automatically detect epistemological bias using a fine-tuned tagger and a lexicon lookup. The result demonstrated 10 article headlines of Meghan Markle and 10 articles of Kate Middleton through the framework. In essence, journalists can use the system to detect potential bias types and injustices that might be present in their headlines before publishing.

**Summary Of The Review:**

This is a good paper and if the weaknesses are addressed, it will show good reproducibility and would benefit the community immensely in other domains away from the news media.

---

> ### Author Response · Authors · 2022-11-16
> **Reply 1: zugw**
>
> Thank you for your review. We have addressed your concerns with edits to our paper. We welcome additional feedback and improvements.
> - **Thank you for highlighting that our approach is current. We also would like to note that we have updated sections 2, 3, 3.1, 3.2, and 4.1 to be more detailed. We have also updated sections 1, 3.1, and 4.2 for clarity.**
> - *"This is a good paper and if the weaknesses are addressed, it will show good reproducibility and would benefit the community immensely in other domains away from the news media."*  > **Thank you for this positive feedback. We have addressed all of the mentioned weaknesses and have further emphasized the positive strengths of the paper. We have updated sections 1 and 4.3 to discuss some other domains in which the framework can be used. We also updated the Title and abstract to emphasize the news media is only a use case for this work.**
> - *"The framework developed was not shown - a schematic diagram or an algorithm would have sufficed."* > **We have updated the paper to include a diagram of the framework in Section 3.**
> - *"The tagger-UI was not presented in the work - a screenshot would have sufficed"* > **We have updated the paper to include 2 screenshots of the UI in Section 3.4 and in the Appendix.**
> - *"No mathematical model was demonstrated to prove the theory behind the entire process."* > **We have clarified our contribution #4 in section 1 and section 5 to make it more clear of our goal to provide empirical evidence as a justification for using our framework to automatically detect biases and injustices. Also to add some mathematical support we have updated sections 3.1, 3.2, and 4.1.**
> - *"How the sentiments (pos, neg) were derived was not shown clearly."*  > **We have updated the paper to show how the sentiments were achieved in the Appendix when discussing Table 6.**
>
> We thank you for your review and welcome any additional feedback and further review. You will find our updated paper is now uploaded. We thank you for your support of our paper.

---

### Official Review · Reviewer_VEMe · 2022-10-25

**Confidence:** 4
**Correctness:** 2
**Technical Novelty And Significance:** 1
**Empirical Novelty And Significance:** 1
**Recommendation:** 3

**Clarity, Quality, Novelty And Reproducibility:**

- The work used previous existing models. But there are no details to provide justification about why the models are used (i.e., whether using the models are good choices) and how accurate they are.


**Strength And Weaknesses:**

Strengths:
- I find the topic is interesting.
- The paper throws big questions (in Sec 2), e.g., what kinds of implications does our use of these words impact society?
- It shows how the pipeline is from the input text to the output.

Weaknesses:
- The paper doesn't describe the motivation in a convincing way.
- The technical details are missing. It is difficult to know how the actual detection part works unless we read the previous papers.
- There is no discussion about the accuracy of those models.
- It seems that the propose pipeline uses existing models (with a small modification in the first model, which I was not really convinced of), and the interactive interface is a contribution. However, there is no screenshot of or an emphasis on the interface. The paper just describes what it does, and that's all.

Detailed comments are below. Major and minor comments are mixed.
- use citations in a right format for readability. In addition, there are many incomplete sentences.
- "give journalist and editors a tool which will help them learn" <= They are experts, and they sometimes/often use the biased words intentionally to present their "opinions" and influence the public. In this case, how could your tool be helpful?
- The paper seems to assume that the existing models give correct output, and there is no discussion about the accuracy of those models.
- "it is difficult for common people to identify a biased word" <= Then, how was the training data for the existing models created? Was there any training involved?
- "difficultly" => difficulty
- expert linguistic and epistemological bias features in the tagger model<= Were these features manually extracted/annotated by experts? what kinds of features are they?
- When you extend the model by including TMI, is this also manual feature extraction?
- I'm confused with the TMI features because they look like they are already indicating biases, i.e., the output. So, the input are bias features, and the output are the probabilities indicating biases? It feels like the features are part of the intended output.
- Why did you make a debiasing weight 1.3 as opposed to previous work? Any particular reason?
- In 3.2, "We rank these outputs based on their semantic similarity to the headlines." <= what is the semantic similarity between streotypes and new headlines? I'm not sure if the term "semantic similarity" is correct here. "The stereotype and stereotype concept most correlated with the sentence" <= how do you compute this?
- Does the tagger model take into account contexts where a word is used? How is the output of this model different from using the epistemological lexicon directly to find biased words? It seems that the words that are not in this lexicon are not included for the study (the output) anyway.
- interactive interface seems like the biggest contribution of this paper, but no screenshot or any study regarding that?
- Table 1 uses a lot of space, but I'm not sure what the point is here showing all those. This space should be used for describing more technical details.
- need a brief description of "Meghan Markle and Kate Middleton". not all the readers know them.
- data: Is the data of Mechan Markle and Kate Middleton from the bias data corpus (Pryzant et al., 2020b)? How many headlines? <= Section 6 says it was only 20. How did you get this data?
- "if the measures and metrics suggested here become targets, they will cease to be useful." <= what does this mean?
- I think it would be more helpful to design/interpret the current comparative study to show interesting/meaningful findings (e.g., in social sciences).

**Summary Of The Paper:**

This paper proposes a pipeline that detects epistemologically biased words (e.g., ripped - negative) and stereotypes (e.g., women should be dressed like brides) in texts. The authors develop an interactive user-interface to show the detected output with a goal of helping journalists and editors. The backend of this interface (i.e., the detection part) consists of three previous models. The paper shows a study about two subjects, Meghan Markle and Kate Middleton, demonstrating the output from the system about them.

**Summary Of The Review:**

Although the topic of the paper is very interesting and important, I think the paper needs more work - little technical contribution and no deeper discussion/study about the efficacy of the proposed interactive interface. In addition, no technical details are provided to let readers understand how those biased words are detected.

---

> ### Author Response · Authors · 2022-11-16
> **Reply 1: VEMe**
>
> Thank you for your review. We have addressed your concerns with edits to our paper. Additional feedbacks are welcomed.
>
> - Thank you for the positive feedback of *"It shows how the pipeline is from the input text to the output."* > **We have emphasized this point by adding an image of the framework to section 3.**
> - *"The paper doesn't describe the motivation in a convincing way."* > **We have added more explanation of the importance of detecting these injustices in section 1 and have previously illustrated some findings that can be extracted from this work in section 4.2**
> - *"The paper seems to assume that the existing models give correct output, and there is no discussion about the accuracy of those models."* > **To address this concern we have added an ablation study in section 4.1. We also added accuracy results from the SBF and CoStar models in section 3.2.**
> - expert linguistic and epistemological bias features in the tagger model *"Were these features manually extracted/annotated by experts? what kinds of features are they?"* > **We have updated section 3.1 to answer this question.**
> - *"there is no screenshot of or an emphasis on the interface."* > **We have now added screenshot of the UI to section 3.4 and in the Appendix.**
> - "give journalist and editors a tool which will help them learn" *"They are experts, and they sometimes/often use the biased words intentionally to present their "opinions" and influence the public. In this case, how could your tool be helpful?"* > **We have updated section 4.3 to answer this question.**
> - "it is difficult for common people to identify a biased word" *"Then, how was the training data for the existing models created? Was there any training involved?"* > **We have updated sections 2 and 3.5 to answer this question.**
> - *"When you extend the model by including TMI, is this also manual feature extraction?"* > **No, it is not a manual feature extraction. Based on our hypothesis of TMI as defined in section 3.1 of our paper, we created methods in python which makes use of the core NLP dependency tree and a tree traversal algorithm to go through the sentence starting from the root node of the sentence, then count the number of adjectives and adverbs in the sentence.**
> - *"I'm confused with the TMI features because they look like they are already indicating biases, i.e., the output. So, the input are bias features, and the output are the probabilities indicating biases? It feels like the features are part of the intended output."* > **We have updated section 3.1 to answer this question.**
> - In 3.2, "We rank these outputs based on their semantic similarity to the headlines." *"what is the semantic similarity between stereotypes and new headlines? I'm not sure if the term "semantic similarity" is correct here."* "The stereotype and stereotype concept most correlated with the sentence" *"how do you compute this?"* > **We have updated section 3.2 to answer this question.**
> - *"Does the tagger model take into account contexts where a word is used? How is the output of this model different from using the epistemological lexicon directly to find biased words? It seems that the words that are not in this lexicon are not included for the study (the output) anyway."* > **Yes, it takes into account contexts. We have updated section 3.5 to expand**
> - *"need a brief description of "Meghan Markle and Kate Middleton". not all the readers know them."* > **We have added a briefing in section 4.2.**
> - *"data: Is the data of Meghan Markle and Kate Middleton from the bias data corpus (Pryzant et al., 2020b)? How many headlines? <= Section 6 says it was only 20. How did you get this data?"* > **We previously discussed this in Section 4.2 and have updated it to expand the point further.**
> - "if the measures and metrics suggested here become targets, they will cease to be useful." *"what does this mean?"* > **We have expanded on this quote for our contexts in section 4.3**
> - *"I think it would be more helpful to design/interpret the current comparative study to show interesting/meaningful findings (e.g., in social sciences)."* > **We previously mentioned the findings that come from the study and the implications of the findings in the last paragraph of section 4.2.**
> - In reference to the debiasing weight > **We have misspoken here: 1.3 was the suggested debiasing weight to use when training the detection model - although 1.0 is the default value originally set from the previous work. We have removed this from the paper to mitigate the confusion.**
> - In reference to citations, grammar, and spelling: **we have changed all of the citations to be more readable and fixed spelling errors and grammar errors throughout the paper. We have also fixed spacing issues.**
>
> We thank you for your review and welcome any additional feedback and further review. You will find our updated paper is now uploaded. We ask you to review these updates and consider increasing your support for our paper.

---

> ### Author Response · Authors · 2022-11-17
> **We have addressed your concerns regarding the technical details of the detection part**
>
> - *"The technical details are missing. It is difficult to know how the actual detection part works unless we read the previous papers."* >>> **we have updated the paper to add more technical details about the detection model in sections 3.1, 4.1 and in the Appendix A.**

---

### Official Review · Reviewer_gnJX · 2022-10-27

**Confidence:** 3
**Correctness:** 4
**Technical Novelty And Significance:** 2
**Empirical Novelty And Significance:** 2
**Recommendation:** 5

**Clarity, Quality, Novelty And Reproducibility:**

While the end to end proposal of framework is interesting, the technical contributions can be strengthened. I also felt that the details about using Co-STAR models and SBF models could be more elaborative making it easier to reproduce the results.

**Strength And Weaknesses:**

Strengths:
1. Authors tackle an important and interesting problem
2. They propose an approach with a real-world application in mind and propose a framework that can be used in production to detect bias in text

Weakness:

1. I fail to see why the methodology has not been proposed as a generic approach to detect epistemological bias but has been restricted to news media.
2. The problem that the authors are trying to capture is slightly confusing. Are they trying to detect epistemological biases? Are they trying to detect injustice in text? How are the two different?
3. Authors mention detecting TMI but I couldn't find the process that they are using for the same
4. The comparative test seems to have a scope of only 20 headlines which doesn't seem enough

**Summary Of The Paper:**

Authors propose a framework to detect epistemological biases and stereotypes in text and use this to assist with detecting injustice in text. They fine-tune a BERT based model for bias detection and Co-STAR and Social Bias Frames to detect stereotypes and stereotype concepts. They also propose an interactive interface that can be used by editors and journalists to learn about potential bias in the news media text they are publishing.



**Summary Of The Review:**

Please see the sections above

---

> ### Author Response · Authors · 2022-11-16
> **Reply 1: gnJX**
>
> Thank you for your review. We have updated our paper to address these concerns. We welcome additional feedback and improvements.
>
> - Thank you for the positive feedback of *"They propose an approach with a real-world application in mind and propose a framework that can be used in production to detect bias in text."* > **We have further emphasized this point by updating sections 1, 3, 4.2, and 4.3.**
> - *"I fail to see why the methodology has not been proposed as a generic approach to detect epistemological bias but has been restricted to news media."* > **We have updated sections 1 and 6 of the paper to further highlight that using the news media is just one use case option by mentioning some other domains in which the framework will be helpful. We have also discussed this in section 4.3. We have also updated the Title of the paper and the Abstract to make this more clear for readers.**
> - *"The problem that the authors are trying to capture is slightly confusing. Are they trying to detect epistemological biases? Are they trying to detect injustice in text? How are the two different?"* > **Bias leads to injustices and our aim was to show the automatic detection of biases and also show how those biases could cause injustice. We highlight this in sections 1, 2, and 4.2. We have updated the paper to further emphasize this point in sections 1 and 5.**
> - *"Authors mention detecting TMI but I couldn't find the process that they are using for the same."* > **We have added more details about TMI in sections 3.1 and 4.1 to address this.**
> - *"The comparative test seems to have a scope of only 20 headlines which doesn't seem enough"* > **We acknowledged the limitation of using a few samples of headlines in our evidence and analysis portion to show the task of detecting injustice in Section 6. We would like to highlight when using the tool it can scale up to be used on large-scale media data. It can span large-scale, multiple volumes of news articles which can be collected with an API crawl, stored in a document, and uploaded. We have updated our paper to show this by including the process by which one might add multiple volumes of data in section 3. Please also see the added screenshots of the UI in section 3.4 and the Appendix. Additionally, we have updated the paper to mention other domains in which the tool could be used in section 1.**
> - *"While the end to end proposal of framework is interesting, the technical contributions can be strengthened"* > **We have updated the paper to add more technical details about the models and framework in sections 3, 3.1, 3.2, and 4.1. Another technical contribution is the creation of the UI, which we have updated the paper to add screenshots of the UI in section 3.4 and the Appendix. Our final technical contribution is the fine-tuning of the Tagger Model. We have updated sections 3.1 and 4.1 with more technical details about this.**
> - *"I also felt that the details about using Co-STAR models and SBF models could be more elaborative making it easier to reproduce the results"*  > **We have updated Section 3.2 to address this concern. We have also updated section 7 to include the GitHub link for reproducing the CO-STAR and SBF models.**
>
> We thank you for your review and welcome any additional feedback and further review. You will find our updated paper is now uploaded. We ask you to kindly review these updates and consider increasing your support for our paper.

---

### Author Response · Authors · 2022-11-16
**Title and Abstract Slight Update**

We have updated our Title and Abstract (with minimal changes) to address the concerns of reviewers. The updated paper is now uploaded and reflects these changes.

---

### Decision · Program_Chairs · 2023-01-20

**Decision:**

Reject

**Justification For Why Not Higher Score:**

N/A

**Justification For Why Not Lower Score:**

N/A

**Metareview: Summary, Strengths And Weaknesses:**

This work proposes a combined model to detect epistemological biases and stereotypes in text. Reviewers agreed this work tackles an important and interesting question, with explicit limitation discussion and an interactive interface. On the other side, reviewers questioned the technical contribution given much detail is missing, and found the quantitative part weak. There are also concerns about motivation and how this tool could be used more broadly and quantitatively on social media. After reviewing the strengths, weaknesses, and authors' responses, I recommend not accepting the current version and encouraging the authors to improve it in accordance with the reviewers' recommendations.


**Summary Of Ac-Reviewer Meeting:**

N/A